# PirAB^VP^ Toxin Binds to Epithelial Cells of the Digestive Tract and Produce Pathognomonic AHPND Lesions in Germ-Free Brine Shrimp

**DOI:** 10.3390/toxins11120717

**Published:** 2019-12-09

**Authors:** Vikash Kumar, Lobke De Bels, Liesbeth Couck, Kartik Baruah, Peter Bossier, Wim Van den Broeck

**Affiliations:** 1Lab of Aquaculture & Artemia Reference Center, Department of Animal Sciences and Aquatic Ecology, Faculty of Bioscience Engineering, Ghent University, 9000 Ghent, Belgium; Peter.Bossier@ugent.be; 2ICAR-Central Inland Fisheries Research Institute (CIFRI), Barrackpore 700120, India; 3Department of Morphology, Faculty of Veterinary Medicine, Ghent University, Salisburylaan 133, 9820 Merelbeke, Belgium; lobke.debels@ugent.be (L.D.B.); liesbeth.couck@ugent.be (L.C.); Wim.VandenBroeck@UGent.be (W.V.d.B.); 4Department of Animal Nutrition and Management, Faculty of Veterinary Medicine and Animal Sciences, Swedish University of Agricultural Sciences, 75007 Uppsala, Sweden; kartik.baruah@slu.se

**Keywords:** PirAB^VP^ toxins, brine shrimp, digestive tract, sloughing, epithelial cells, necrosis

## Abstract

Acute hepatopancreatic necrosis disease (AHPND), a newly emergent farmed penaeid shrimp bacterial disease originally known as early mortality syndrome (EMS), is causing havoc in the shrimp industry. The causative agent of AHPND was found to be a specific strain of bacteria, e.g., *Vibrio* and *Shewanella* sps., that contains pVA1 plasmid (63–70 kb) encoding the binary PirA^VP^ and PirB^VP^ toxins. The PirAB^VP^ and toxins are the primary virulence factors of AHPND-causing bacteria that mediates AHPND and mortality in shrimp. Hence, in this study using a germ-free brine shrimp model system, we evaluated the PirAB^VP^ toxin-mediated infection process at cellular level, including toxin attachment and subsequent toxin-induced damage to the digestive tract. The results showed that, PirAB^VP^ toxin binds to epithelial cells of the digestive tract of brine shrimp larvae and produces characteristic symptoms of AHPND. In the PirAB^VP^-challenged brine shrimp larvae, shedding or sloughing of enterocytes in the midgut and hindgut regions was regularly visualized, and the intestinal lumen was filled with moderately electron-dense cells of variable shapes and sizes. In addition, the observed cellular debris in the intestinal lumen of the digestive tract was found to be of epithelial cell origin. The detailed morphology of the digestive tract demonstrates further that the PirAB^VP^ toxin challenge produces focal to extensive necrosis and damages epithelial cells in the midgut and hindgut regions, resulting in pyknosis, cell vacuolisation, and mitochondrial and rough endoplasmic reticulum (RER) damage to different degrees. Taken together, our study provides substantial evidence that PirAB^VP^ toxins bind to the digestive tract of brine shrimp larvae and seem to be responsible for generating characteristic AHPND lesions and damaging enterocytes in the midgut and hindgut regions.

## 1. Introduction

The outbreak of acute hepatopancreatic necrosis disease (AHPND) caused by *Vibrio* spp. has been particularly devastating in the cultivation of shrimp in a number of countries [1,2,3,4,5]. The shrimp production in AHPND-affected regions has at times dropped considerably (to ~60%) and disease has caused an estimated US $43 billion loss across Asia (China, Malaysia, Thailand, Vietnam) and in Mexico in last 10 years [3,6,7]. The *Vibrio* spp. becomes virulent by acquiring a 63–70 kb plasmid (pVA1) encoding the binary PirAB^VP^ toxins, which consist of two subunits PirA^VP^ and PirB^VP^, and is homologous to the *Photorhabdus luminescens* insect-related (Pir) toxins PirA/PirB [8,9]. The PirAB^VP^ toxins are the primary virulence factor of AHPND-causing bacteria that mediates AHPND and mortality in shrimp [10]. The binary PirAB^VP^ toxins mainly target the hepatopancreas (digestive gland) of shrimp and damage the R (resorptive), B (blister), F (fibrillar), and E (embryonic) cells, resulting in dysfunction and massive mortalities (up to 100%) within 20–30 days of shrimp post-larvae stocking [2,5,11]. Since the impact of these binary toxins are significant in shrimp aquaculture, more research attention is needed to unravel the toxin-mediated infection process at cellular level.

Among the binary PirAB^VP^ toxins, PirA^VP^ facilitates target-specific recognition of toxins by binding to certain ligands on the cell membrane and receptors (e.g., monosaccharides like N-acetylgalactosamine (GalNAC) and oligosaccharides), while the PirB^VP^ toxin (containing N-terminal domain, PirBN and C-terminal domain, PirBC), is mainly responsible for cell death via pore formation, and is involved in protein–protein and protein–ligand interactions [3,12,13]. Moreover, together PirA^VP^ and PirB^VP^ toxins form a complex and act synergistically, resulting in increased toxicity of PirAB^VP^ toxins on the experimental animals [9,13].

In this study, using a highly controlled gnotobiotic brine shrimp model system, we aimed to investigate the morphological changes in the guts of germ-free brine shrimp larvae during PirAB^VP^ toxin challenge. Furthermore, we also unraveled that PirAB^VP^ toxins bind to epithelial cells of the digestive tract, induce necrosis, and damage the cellular structure, including the nucleus, mitochondria, junctional complex, rough endoplasmic reticulum (RER), etc., which leads to the subsequent death of challenged brine shrimp larvae. The knowledge gained from this study will facilitate future research which aims at the comparison of the digestive tract morphology after the introduction of anti-AHPND therapy in the culture system.

## 2. Results

### 2.1. PirAB Toxin Binds the Digestive Tract and Induces Sloughing of Epithelial Cells in Brine Shrimp (Artemia franciscana) Larvae

Immunohistochemistry using Mab (monoclonal antibody) specific to His_6_-tagged PirAB^VP^ toxins, showed strong immunoreactivity in the epithelium of digestive tract from PirAB^VP^-challenged brine shrimp larvae. The PirAB^VP^ immunoreactivity was seen from 12 h post-challenge in close contact with the brush border of the enterocytes (Figure 1C–L). In the intestinal lumen, moderately electron-dense cells of variable shapes and size were observed 12 h post-challenge. Shedding or sloughing of enterocytes in the midgut and hindgut regions was regularly visualized from 12 h post challenge onwards until the end of the experiment (60 h post-challenge) (Figure 1C–L). After 60 h post-challenge, the epithelium was severely damaged in the challenged brine shrimp larvae (Figure 1K,L). Additionally, the remaining cellular components, such as the pyknotic nuclei and lysed cellular membrane, were further detached into the lumen and showed signs of degeneration. Moreover, the digestive tract epithelial cells in the control larvae had a normal appearance with the presence of the nucleus, brush border, and prominent gut lumen (Figure 1A,B). In addition, the Mab specific to the His_6_-tagged PirAB^VP^ toxins bound only to PirAB^VP^ toxins and did not show any cross-reactivity to other endogenous cellular histidine.

### 2.2. PirAB Toxin Induced Accumulation of Cellular Debris in the Digestive Tract Is of Epithelial Cell Origin

To further confirm that PirAB^VP^-induced accumulation of cellular debris in the digestive tracts of brine shrimp contains cells of epithelial origin, cytokeratin-specific polyclonal antibody (Pab) was used to localize the epithelial cells in intestinal lumen. Since the sloughing of enterocytes was observed 12 h post-challenge onwards, 12 and 24 h post challenge brine shrimp larvae samples were used to visualize the changes. The PirAB^VP^ toxin-challenged brine shrimp larvae sections treated with anti-keratin antibody (cytokeratin Pab), which form the intermediate-sized filaments in the cytoplasm of epithelial cells [14], gave strong immunoreactive staining in the intestinal lumen, indicating positive signals for epithelial cells in the cellular debris. The cytokeratin Pab immunoreactive signals was observed in the midgut and hindgut of the digestive tract in brine shrimp larvae samples at 12 and 24 h post toxin challenge (Figure 2C–H). Moreover, in the control group larvae (no PirAB^VP^ toxin) at 12 and 24 h post experiment, the digestive tract appeared normal, with no accumulation of cellular debris (Figure 2A,B).

### 2.3. PirAB Toxin Binding Leads to Damage and Necrosis of the Digestive Tract in Brine Shrimp Larvae

Next, to confirm that PirAB^VP^ toxins are associated with damage on the digestive tracts of brine shrimp larvae, electron microscopical analysis was performed in the 48 h post-challenge brine shrimp larvae samples. TEM ultrastructural observations showed that digestive tract epithelium composing the enterocytes were damaged to different degrees, with focal to extensive necrosis, in PirAB^VP^ toxin-challenged brine shrimp larvae (Figure 3A–D). Furthermore, the analysis showed that epithelial enterocytes in the midgut and hindgut regions were damaged showing nuclear pyknosis, cell vacuolisation, and mitochondrial and rough endoplasmic reticulum (RER) damage to different degrees (Figure 3A–D). Notably, there were rod-shaped bacteria observed in the digestive tracts inside and outside the intestinal lumen of PirAB^VP^ toxin-challenged brine shrimp larvae. Since germ-free brine shrimp larvae were used for the analysis, these bacteria were identified as the autoclaved bacterial feed given during the challenge experiment. Apparently, the larvae were unable to digest them because of damage in the digestive tract. Moreover, in the control brine shrimp larvae group, the morphology of the digestive tract epithelial enterocytes appeared normal, with an intact mitochondrion, nucleus, rough endoplasmic reticulum (RER), and intercellular junctions (Figure 4A–F). The data indicated that damage of the digestive tract and accumulation of cellular debris in the intestinal lumen was due to the PirAB^VP^ toxin-challenge, which leads to subsequent death of challenged larvae as reported in our previous study [9].

## 3. Discussion

Acute hepatopancreatic necrosis disease (AHPND), earlier named as early mortality syndrome (EMS), has been particularly devastating in the cultivation of shrimp, causing massive mortality (up to 100%) within 20–30 days of post-larvae stocking [5,11,15]. Since the AHPND outbreak first appeared in China in 2009, it has spread to Vietnam (2010), Malaysia (2011), Thailand (2012), Mexico (2013), Philippines (2015), South America (2016), Australia (2016), Bangladesh (2019), and USA (2019) [7,8,16,17,18,19]. The causative agent of AHPND was found to be a specific strain of bacteria, including *Vibrio parahaemolyticus*, *V. punensis*, *V. harveyi*, *V. owensii*, *V. campbelli*, and *Shewanella* sp. that contains pVA1 plasmid (63–70 kb), encoding the binary PirA^VP^ and PirB^VP^ toxins [9,20,21,22,23]. Although, the PirAB^VP^ toxins were reported as the primary virulence factor of AHPND-causing bacteria that mediates AHPND and mortality in shrimps [9,13], the toxin-mediated infection process at the cellular level still remains unclear. To this end, brine shrimp (*Artemia franciscana*), an aquatic invertebrate crustacean and a model for crustacean shrimp [24,25,26] was used to develop an infection model for AHPND and investigate the morphological changes in the guts of germ-free brine shrimp larvae during the PirAB^VP^ toxin challenge. Subsequent studies were carried out to visualize the PirAB^VP^ toxin-binding and early pathogenesis in the digestive tracts of brine shrimp larvae, to evaluate the host–pathogen interaction.

The Photorhabdus insect-related (Pir) toxins were first identified in *Photorhabdus luminescens*, a bacterium that maintains a symbiotic relationship with entomopathogenic nematodes of the family Heterorhabditidae [27,28]. The PirAB toxins act as binary proteins; they are encoded by the PirA and PirB genes, and both proteins are necessary for oral toxicity in moths and mosquitoes [29,30]. The pathology of Pir oral toxicity in larvae of the moth *Plutella xylostella* is in the midgut epithelium, and its toxicity results in swelling and shedding of the (apical) epithelial cells [31]. In shrimp aquaculture, the virulent AHPND-causing bacteria containing pVA1 plasmid encodes PirAB^VP^ toxin genes, homologous to the insecticidal PirA/PirB toxins’ genes, were absent in all non-AHPND bacterial species [32]. Moreover, as shrimp and insects are both arthropods, the AHPND-affected shrimp have pathological responses similar to the PirAB midgut toxicity in insects [5]. In this study, using the germ-free brine shrimp model system, it was shown that PirAB^VP^ toxins bind to gut epithelial tissue, especially midgut and hindgut epithelium and produce pathognomonic AHPND lesions. The study further demonstrates that PirAB^VP^ toxins binding induce necrosis and damages the cellular structure of the digestive tract including nucleus, mitochondria, rough endoplasmic reticulum (RER), etc. in the challenged brine shrimp larvae.

It was reported previously that the PirA^VP^ and PirB^VP^ toxins both form a three-domain complex, and act synergistically to produce the characteristic symptoms of AHPND in the shrimp [3,33]. In addition, PirAB^VP^ toxins have been reported to induce AHPND, possibly via recognition and binding to certain ligands on the cell membrane and receptor, oligomerization, and pore formation on the cell membrane, that leads to cell death [13]. Our results indicate that PirAB^VP^ produce characteristic symptoms of AHPND in brine shrimp larvae. In the PirAB^VP^-challenged brine shrimp larvae, shedding or sloughing of epithelial enterocytes in the midgut and hindgut regions was regularly visualized, and the intestinal lumen was filled with moderately electron-dense cells of variable shapes and sizes (Figure 1C–L). Additionally, as the infection proceeded, the epithelium was severely damaged in the challenged brine shrimp larvae at 60 h post-challenge, and the remaining cellular components were further detached into the lumen, and showed signs of degeneration, such as pyknotic nuclei and lysed cellular membranes (Figure 1K,L). Moreover, the previous studies have documented the specificity of PirAB^VP^ toxins attachment, in which PirAB^VP^ toxins were reported to bind with epithelial cells of hepatopancreas and induce AHPND in shrimp [3,34]. However, the present study showed that specificity of PirAB^VP^ toxins is not very high, and additionally that the attachment is not hepatopancreas-dependent. The PirAB^VP^ toxins mainly target epithelial cells of the digestive tract and bind with midgut and hindgut epithelium. To the authors’ knowledge, no ultrastructural findings have been published previously which illustrate this type of interaction, in which a toxin binds to the epithelial cells and induces a characteristic AHPND lesion in the digestive tract, which leads to subsequent death of challenged larvae, as reported in our previous study [9]. Thus, this finding is the first of its kind, possibly pointing towards a new insight on the infection process of AHPND-causing PirAB^VP^ toxins. 

However, it is hard to generalize that the accumulation of electron-dense cells of variable shapes and sizes in the intestinal lumen of the challenged larvae were PirAB^VP^-induced and of epithelial cell origin, since autoclaved bacterial feed given during the experiment can accumulate and form the cellular mass. Hence, we tried to localize the epithelial cells in intestinal lumen and target keratin groups of fibrous proteins, which form the structural framework of epithelial cells [19,35]. The results indicate that in the PirAB^VP^-challenged brine shrimp larvae, the observed cellular debris in the intestinal lumen are of epithelial cell origin (Figure 2C–H). These observations further indicate that PirAB^VP^ toxins bind epithelial enterocytes in the midgut and hindgut and induce sloughing that leads to the production of cellular debris in the intestinal lumen.

Since it is evident that PirAB^VP^ toxins produce pathognomonic AHPND lesions and damage the digestive tract of brine shrimp larvae, we further studied the detailed morphology of the digestive tract in brine shrimp larvae [18,36]. The results demonstrate that the PirAB^VP^ toxin-challenge produces focal to extensive necrosis and damages epithelial enterocytes in the midgut and hindgut regions, resulting in nuclear pyknosis, cell vacuolisation, and mitochondrial and rough endoplasmic reticulum (RER) damage to different degrees (Figure 3A–D). In addition, autoclaved rod-shaped bacterial cells given as feed during the challenge assay were observed in the digestive tract, inside and outside the intestinal lumen of the PirAB^VP^ toxin-challenged brine shrimp larvae. As germ-free brine shrimp larvae were used for the analysis, the results indicate that PirAB^VP^ toxins affect the digestive process, and larvae were unable to digest the supplied food. Moreover, the bacterial cells were also observed outside the intestinal lumen, which showed that PirAB^VP^ toxins might activate a Cry-like pore-forming domain [13,33] and induce damage of the digestive tract, resulting in the movement of bacterial cells outside of the intestinal lumen (Figure 3A–D).

## 4. Conclusions

In summary, these data illustrate the PirAB^VP^ toxin-mediated infection process and pathogenesis in brine shrimp larvae. Our findings also imply that PirAB^VP^ toxins seem to be responsible for generating characteristics of AHPND lesions, and damage epithelial enterocytes in the midgut and hindgut regions of the digestive tract. This in vivo visualization is an easy model that might help to study therapies that aim at toxin production interference.

## 5. Materials and Methods

### 5.1. Preparation of Recombinant V. parahaemolyticus PirA^VP^ and PirB^VP^

The PirA^VP^ plasmid (pET21b PirA^VP^) and PirB^VP^ plasmid (pET21b PirB^VP^), obtained from Taiwan, were transformed into *E. coli* Rosetta (DE3) competent cells and the expression of the recombinant, C-terminal His_6_-tagged PirA^VP^ and PirB^VP^ proteins, respectively, were induced by the addition of 0.25 µM of iso-propyl thiogalactoside (IPTG) [3]. After expression had proceeded at 16 °C for 16 h, the recombinant proteins were purified with the magneHis™ protein purification system (Promega Corporation, WI, USA). The purified recombinant PirA^VP^ and PirB^VP^ toxins were dialyzed in phosphate buffer solution (PBS) (Honeywell, Grauwmeer, Belgium) and concentrated with amicon^®^ ultra-15 centrifugal filters (Merck Millipore, Overijse, Belgium).

### 5.2. Detection of Recombinant PirA^VP^ and PirB^VP^ Toxins through SDS-PAGE

Following purification and dialysis, the recombinant PirA^VP^ and PirB^VP^ toxins were collected and immediately preserved at −80 °C for further analysis. Subsequently, the protein was electrophoresed in 4–20% SDS-PAGE gel (BioRad, Belgium), with each lane receiving an equivalent volume (10 µL) of toxin. The gels were then stained with Coomassie Biosafe (BioRad Laboratories) and visualized by a ChemiDoc MP imaging system (BioRad, Nazareth, Belgium) for qualitative analysis of recombinant PirA^VP^ and PirB^VP^ toxins (Figure 5) [9]. The concentration of protein was determined using the Bradford method [37] with bovine serum albumin as the standard.

### 5.3. Axenic Brine Shrimp Hatching

The gnotobiotic brine shrimp larvae was produced by hatching high-quality cysts axenically (germ-free) following decapsulation and hatching procedures as described by Baruah et al. [38]. Briefly, 200 mg of *A*. *franciscana* cysts (EG^®^ type, batch 21452, INVE Aquaculture, Dendermonde, Belgium) were hydrated in 18 mL of distilled water for 1 h. Sterile cysts and larvae were obtained via decapsulation using 660 µL NaOH (32%) and 10 mL NaOCl (50%). During the reaction, 0.2-µm filtered aeration was provided. All manipulation was carried out under a laminar flow hood and all tools were sterilized. The decapsulation was stopped after 2 min by adding 10 mL sterile Na_2_S_2_O_3_ at 10g/L. The decapsulated cysts were washed with filtered autoclaved seawater (FASW) containing 35 g/L of instant ocean^®^ synthetic sea salt (Aquarium Systems, Sarrebourg, France). The cysts were resuspended in a 50 mL tube containing 30 mL FASW and incubated for 24 h on a rotor (4 min^−1^) at 28 °C with constant illumination of approximately 27 µE/m^2^ s. After 28 h of incubation, hatched larvae at developmental stage instar II (mouth is opened to ingest particles) were collected, and the axenicity was verified by spread plating (100 µL) as well as by adding (500 µL) of the hatching water on marine agar and marine broth respectively, followed by incubating at 28 °C for 5 days [39]. Experiments started with non-axenic larvae were discarded.

### 5.4. Brine Shrimp Challenge Assay

The challenge assay was performed according to the method developed by Kumar et al. [8]. Briefly, hatched brine shrimp larvae (at developmental stage II) were collected and group of 10 larvae were transferred to 2 mL sterile eppendorf tubes, containing 1 mL FASW. The earlier study has demonstrated that brine shrimp larvae challenged by immersion with PirA^VP^+PirB^VP^ toxin mixture (1:1 ratio) at 5.2 µg/100 µL, induce significantly high mortality (up to 70%) at 60 h post-exposure [9]. Hence, in this study, the toxic concentration, i.e., 5.2 µg/100 µL, of recombinant PirA^VP^+PirB^VP^ toxin mixture (1:1 ratio) were used and exposed to brine shrimp larvae. After feeding (autoclaved bacteria) and addition of the recombinant PirAB^VP^ toxins, the eppendorf tubes were put back on the rotor and kept at 28 °C. Subsequently, the larvae samples were collected for immunohistochemical analysis at 0, 12, 24, 36, 48, and 60 h after the addition of recombinant toxins. The brine shrimp larvae that were not challenged with recombinant toxins served as a negative control. The assay was performed in triplicates.

### 5.5. Light Microscopical Analysis Using Immunohistochemistry

In total, two separate immunohistochemical strategies were carried out to determine the PirAB^VP^ toxin toxicity in challenged brine shrimp larvae. Briefly, the brine shrimp larvae, from both the treatment and control group, sampled for immunohistochemistry, were fixed for 12 h in a Davidson’s AFA fixative consisting of 80 mL of 100% ethyl alcohol, 15 mL of 40% formaldehyde, and 5 mL of acetic acid, and subsequently transferred to 70% ethanol [40]. After pre-staining with haematoxylin [Haematoxylin (C.I. 75290), Merck KgaA, Darmstadt, Germany], each larva was oriented in 2.0% agarose (electrophoresis grade, 15510-019, Life technologies, Paisley, Scotland) and processed using tissue processor (STP 120, Thermo Scientific, Belgium) for approximately 22 h, followed by embedding in paraffin using an embedding station (EC350-1 and 350-2, Microm international) [41]. Per treatment and per sampling point, 6 brine shrimp larvae were cut into serial sections (dorsoventral, sagittal, and transverse) of 5 µm thickness using a microtome (HM360, Microm international). Subsequently, the histological sections were collected on 3-aminopropyltriethoxysilane (APES) coated slides and placed onto hot plate (60 °C) for 1 h. Afterwards, the slides were transferred and incubated overnight at 37 °C. The tissue slides were cleared in xylene, hydrated in graded ethanol series and epitope retrieval was performed by heating the slides in citrate buffer for 2.5 min at 700–800 W, and for 10 min at 160 W, followed by 30 min incubation at 4 °C. In the first analysis, the consecutive sections were processed for indirect immunoperoxidase staining using anti-His monoclonal antibody 2D5 (Mab) (Bioss antibodies, USA) specific to His_6_-tagged PirAB^VP^ toxins at 1:1000 dilution, with PBS on each consecutive section. Hydrogen peroxide (Dako REAL peroxidase-blocking solution, S2023) was added to the section (to block the peroxidase activity in tissue) for 5 min, and later the slides were incubated with Dako envision (anti-mouse HRP labelled) (Agilent, Lexington, MA, USA) for 30 min. The peroxidase activity was visualized by incubation with 0.03% DAB (Agilent, Lexington, MA, USA) for 5 min. The tissue slides were counterstained with haematoxylin, dehydrated in graded ethanol series, cleared in xylene, and mounted in permount. Tissue slides were analyzed light microscopically (BX61, Olympus) and positive reactions were seen as brown coloration.

In the second analysis, the PirAB^VP^-induced lesions were investigated for epithelial cell origin. The paraffin-embedded histological sections were cleared in xylene, hydrated in graded ethanol series, and transferred to citrate buffer to boil the tissue for 2.5 min at 700–800 W, 10 min at 160 W, followed by 30 min incubation at 4 °C. The tissue sections were incubated with 30% rabbit serum (blocking) for 30 min. Afterwards, the slides were processed for indirect immunoperoxidase staining using polyclonal anti-cytokeratin 20 antibodies (Pab) (SPM191) (Santa Cruz Biotechnology, USA) at 1:100 dilution with antibody diluent. Later, the sections were incubated with hydrogen peroxide (Dako REAL peroxidase-blocking solution, S2023) for 5 min followed by Dako envision (anti-mouse HRP labelled) (Agilent, Lexington, MA, USA) for 30 min. The peroxidase activity was visualized by incubation with 0.03% DAB (Agilent, Lexington, MA, USA) for 5 min. The preparations were counterstained with haematoxylin, dehydrated in graded ethanol series, cleared in xylene, mounted in permount, and analyzed light microscopically (BX61, Olympus). Similarly, the positive immuno-reactivity was visualized as brown coloration.

### 5.6. Transmission Electron Microscopical (TEM) Analysis

The brine shrimp larvae, either challenged (PirAB^VP^ toxin) or not-challenged, were fixed in Karnovsky’s fixative for TEM analysis. After overnight fixation at 4 °C, the specimens were washed in sodium cacodylate buffer (pH 7.4) and post fixed for 1 h 30 min in 1% osmium tetroxide. Subsequently, the samples were dehydrated using Leica tissue processor (EM TP, type 709202). After polymerization, the samples got trimmed with the Leica EM TRIM (S4E, type 702601). The semithin sections (1 µm) were cut with glass knives on a Leica Ultramicrotome (EM UC6) and stained with toluidine blue to select and localize the regions of interest. Later, ultrathin sections (90 nm) were cut with a Leica Ultramicrotome (EM UC6) using a diamond knife (DIATOME, ultra 45°; 2.5 mm), and mounted on formvar-coated single slot copper grids (Formvar solution, EMS). The sections were contrasted with uranyl acetate and lead citrate and examined on a transmission electron microscope JEM-1400 Plus (JEOL, Benelux).

## Figures and Tables

**Figure 1 toxins-11-00717-f001:**
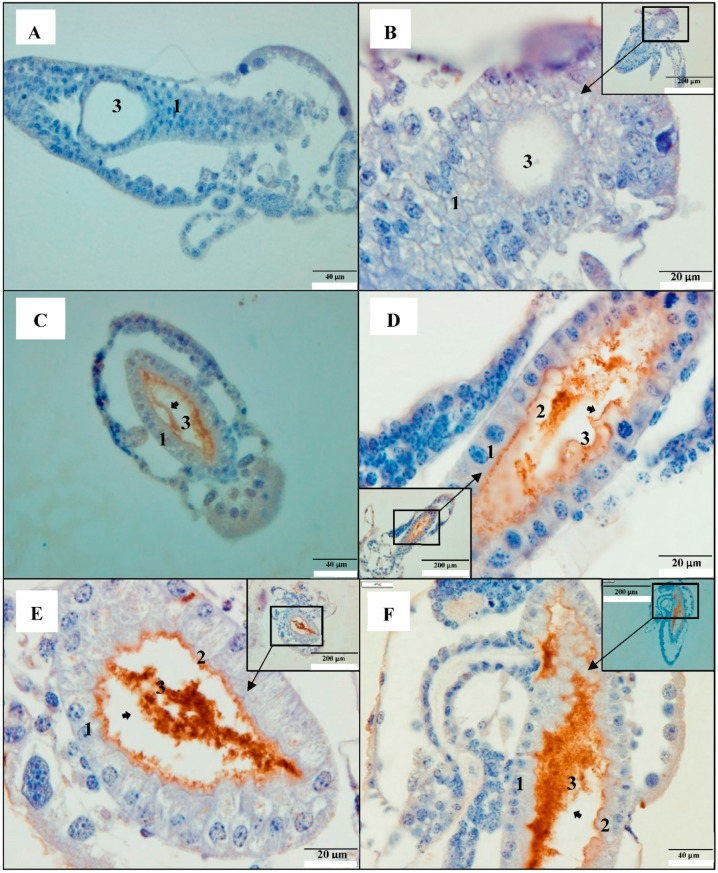
PirAB^VP^ toxins bind to the digestive tract and induce sloughing of epithelial cells in brine shrimp (*Artemia franciscana*) larvae. Immunohistochemistry of brine shrimp (*A. franciscana*) larvae after 0, 12, 24, 36, 48, and 60 h post PirAB^VP^ toxin challenge. The paraffin sections were treated with anti-his Mab (monoclonal antibody) specific to His-tagged PirAB toxins, then counterstained with haematoxylin. Legend: (**1**) Gut cells; (**2**) brush border; (**3**) gut lumen (**A**,**B**): 0 h post-challenge; (**C**,**D**): 12 h post-challenge; (**E**,**F**): 24 h post-challenge (**G**,**H**): 36 h post-challenge; (**I**,**J**): 48 h post-challenge; (**K**,**L**): 60 h post-challenge. (**C**,**L**): PirAB^VP^ toxin binds with digestive tract epithelium and induce shedding or sloughing of enterocytes in the midgut and hindgut regions (**arrowhead**). At 48 h and 60 h post-challenge, epithelium was severely damaged and the remaining cellular components including nuclei were further detached into the lumen and showed signs of degeneration such as pyknotic nuclei and lysed cellular membrane (**arrowhead**).

**Figure 2 toxins-11-00717-f002:**
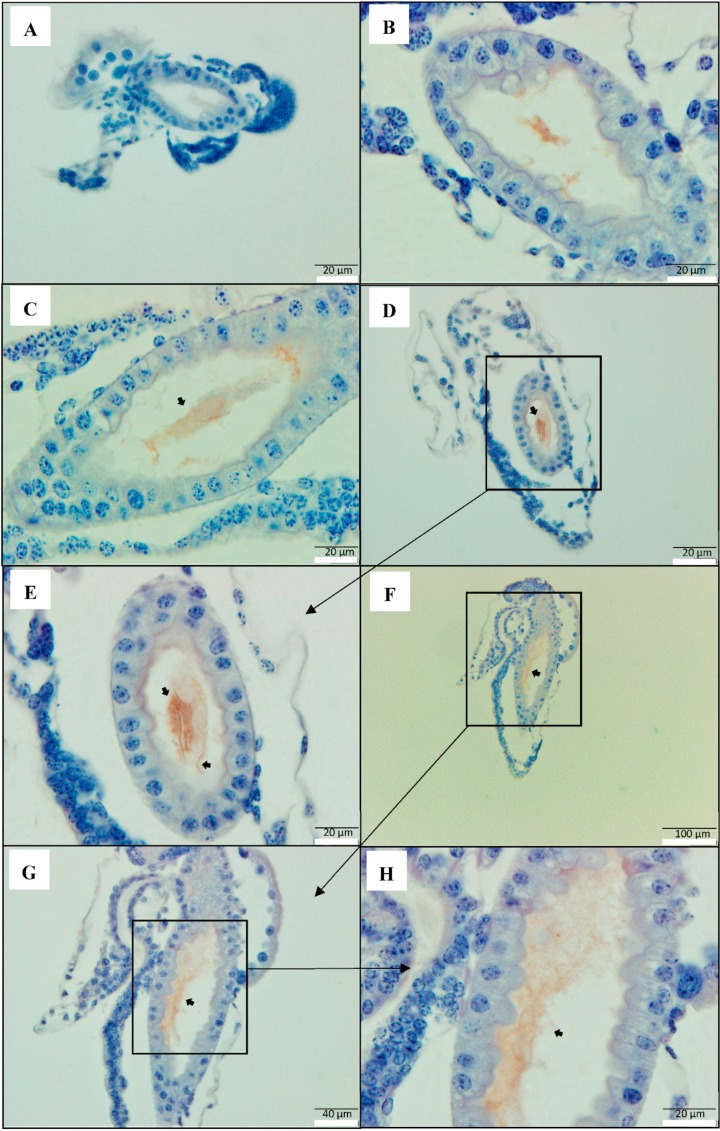
PirAB toxin-induced accumulation of cellular debris in the digestive tract are of epithelial cells origin. Immunohistochemistry of brine shrimp (*A. franciscana*) larvae after 12 and 24 h post PirAB^VP^ toxin challenge. The paraffin sections were treated with anti-cytokeratin polyclonal antibody (Pab), specific to keratin group of fibrous proteins in epithelial cells, then counterstained with haematoxylin. (**A**): 12 h post- experiment, negative control (no PirAB^VP^ toxin); (**B**): 24 h post- experiment, negative control (no PirAB^VP^ toxin); (**C**–**E**): 12 h post-challenge; (**F**–**H**): 24 h post-challenge. (**C**–**H**): PirAB^VP^ challenge-induced sloughing that leads to accumulation of cellular debris in the intestinal lumen (**arrowhead**) is of epithelial cell origin.

**Figure 3 toxins-11-00717-f003:**
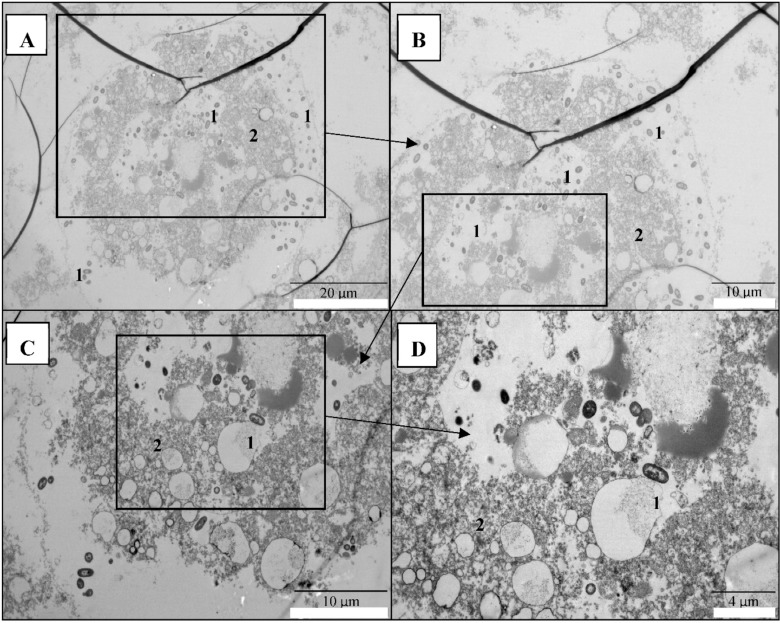
PirAB^VP^ toxin-binding leads to necrosis and damages the digestive tracts of brine shrimp larvae. Transmission electron microscopy (TEM) analysis of brine shrimp (*A. franciscana*) larvae after 48 h post-PirAB^VP^ toxin challenge. (**A**–**D**): PirAB^VP^ toxin challenge damages epithelial enterocytes and induces focal to extensive necrosis, resulting in the movement of bacterial cells (autoclaved feed) outside of the intestinal lumen. Legend: (**1**) Autoclaved bacteria feed for brine shrimp larvae; (**2**) damaged enterocytes with different degrees of focal to extensive necrosis. *Because brine shrimp larvae were too small, before tissue processing and sectioning, the larvae were transferred to 2% agarose which resulted in development of black lines in some TEM figures (**A**,**B**).

**Figure 4 toxins-11-00717-f004:**
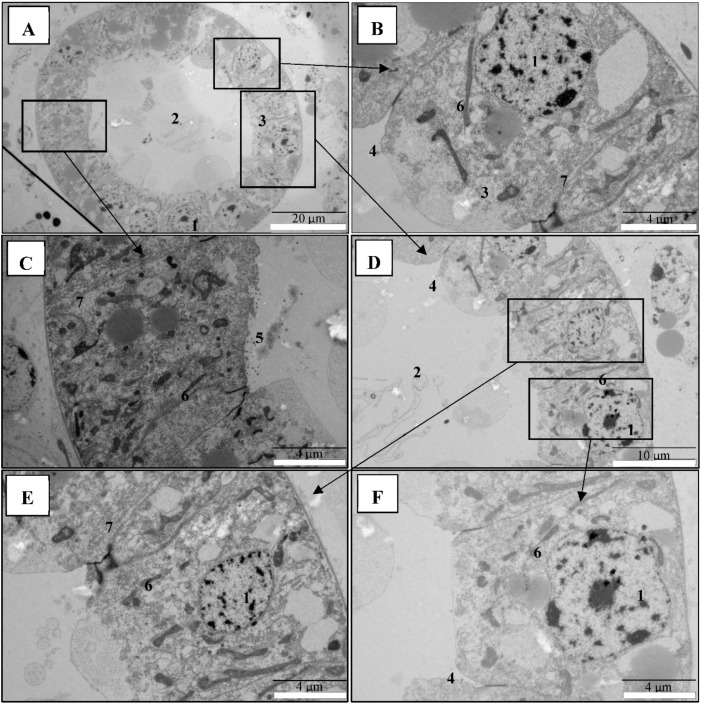
Digestive tract in control brine shrimp larvae appeared normal, with intact cellular membranes. Transmission electron microscopy (TEM) analysis of control group brine shrimp (*A. franciscana*) larvae after 48 h. (**A**–**F**): The digestive tract epithelial enterocytes appeared normal with an intact mitochondrion, nucleus, rough endoplasmic reticulum (RER), and intercellular junctions. Legend: (**1**) nucleus; (**2**) midgut lumen; (**3**) midgut enterocytes; (**4**) tight junction; (**5**) microvilli; (**6**) mitochondria; (**7**) rough endoplasmic reticulum (RER).

**Figure 5 toxins-11-00717-f005:**
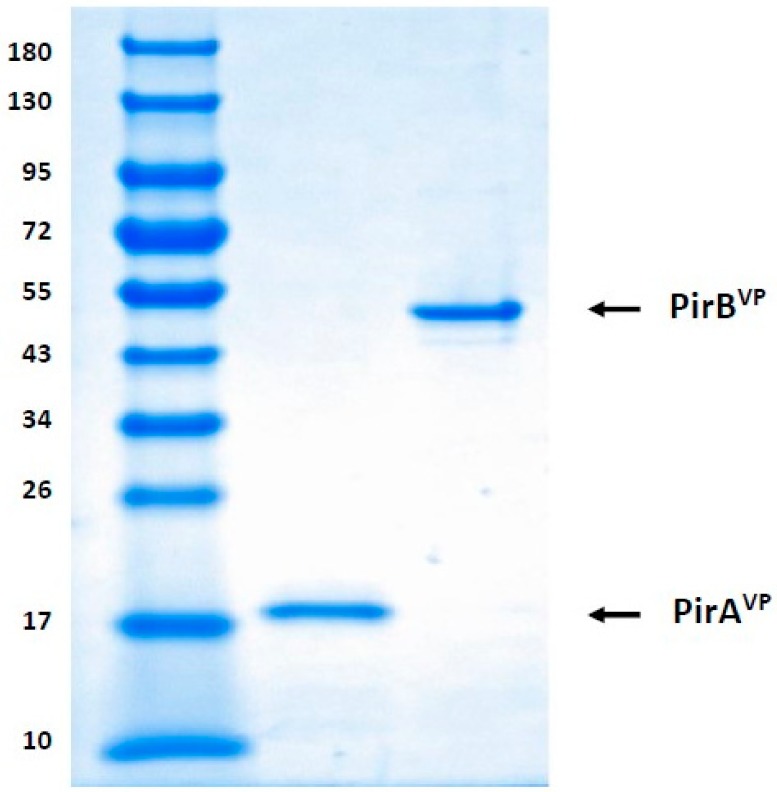
SDS-PAGE analysis of purified *V. parahaemolyticus* PirA^VP^ and PirB^VP^ toxins. Molecular mass standards (M) in kilodaltons (Protein ladder), Lane 1-PirA^VP^ at 13 kDa, Lane 2-PirB^VP^ at 50 kDa.

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
