# Peer review of "PirABVP Toxin Binds to Epithelial Cells of the Digestive Tract and Produce Pathognomonic AHPND Lesions in Germ-Free Brine Shrimp"

_toxins, 2019, doi:10.3390/toxins11120717_

Round 1

Reviewer 1 Report

The manuscript ID: toxins-658720 entitled ‘PirAB toxin binds to epithelial cells of the digestive tract and produce pathognomonic AHPND lesions in germ-free brine shrimp’ evaluated the PirAB toxin-mediated infection process at cellular level, including toxin attachment and subsequent toxin-induced damage to the digestive tract of brine shrimp model system.

These novel results provides substantial evidence that PirAB toxin binds to the digestive tract of brine shrimp larvae and responsible for generating characteristic AHPND lesions and damages enterocytes in the midgut and hindgut regions.

The authors have done important work investigating the production of pathognomonic AHPND lesions in shrimp and imiginizing the damages of enterocytes in the gut.

The writing is good and the data are presented properly in a clear and concise form.

I have several minor concerns in this manuscript.

1) Some words of the bacterial name should be italictized.

2) In the material & method section (5.1.), please clarify the origin of PirA and PirB gene. If the genes were collected from GenBank database, please add the GenBank Nos.

3) In the material & method section (5.1.), please clarify the exact concentration of PirA and PirB, respectively. As I understand, 2.6 μg/100 μl per each subunit?

4) As the toxins were treated in live shrimp, I think the detailed results of mortalities or abnormalities of the shrimp should be presented in the result section.

Author Response

To,

The Reviewer,                                                                                    

Toxins

Manuscript " PirAB toxin binds to epithelial cells of the digestive tract and produce pathognomonic AHPND lesions in germ-free brine shrimp" Manuscript ID: toxins-658720

Dear Reviewer,

We sincerely thank you for your response and comments. We appreciate the thorough review and helpful suggestions, which we believe have contributed to improving the manuscript. We have considered and tried to address all the comments and suggestions given by you.

We have attempted to answer the comments systematically. In the following paragraphs, you will find our response to the comments.

1) Some words of the bacterial name should be italictized.

As suggested, we have carefully checked the entire manuscript and bacterial names were changed to italics

2) In the material & method section (5.1.), please clarify the origin of PirA and PirB gene. If the genes were collected from GenBank database, please add the GenBank Nos.

The PirAVP plasmid (pET21b PirAVP) and PirBVP plasmid (pET21b PirBVP) were obtained from Chu-Fang Lo, National Cheng Kung University, Taiwan. Hence, in the material & method section (5.1.) we have added the description (line 226-227 in the revised paper)

3) In the material & method section (5.1.), please clarify the exact concentration of PirA and PirB, respectively. As I understand, 2.6 μg/100 μl per each subunit?

According to the suggestions, we clearly described the concentration of PirAVP and PirBVP used for challenge assay (Line 264-269 in the revised paper)

4) As the toxins were treated in live shrimp, I think the detailed results of mortalities or abnormalities of the shrimp should be presented in the result section.

Taken account into reviewer comment, we carefully described the rationale of using 5.2 μg/100 μl PirABVP toxin concentration in brine shrimp as model and explained in line 264-269 in the revised paper. Moreover, in our previous study (Kumar et al. 2019. Probing the mechanism of VP AHPND extracellular proteins toxicity purified from Vibrio parahaemolyticus AHPND strain in germ-free Artemia test system. Aquaculture), we have given the overview of PirAB toxin toxicity in different concentration. Hence, in this work we took the reference from previous work and used 5.2 μg/100 μl PirABVP toxin concentration to describe the toxin-mediated cellular changes in Artemia larvae.

Reviewer 2 Report

   This MS describes an image-based study on the cytotoxic effects of Vibrio parahaemolyticus PirAB toxins (PirABvp). By using immunohistochemistry and transmission electron microscopy, authors confirmed PirABvp toxins could induce cell necrosis, which is a pathognomonic AHPND (Acute Hepatopancreatic Necrosis Disease) lesion, by binding to epithelial cells of the digestive tract in germ-free brine shrimp. The results of this MS may provide some useful information in the understanding of AHPND pathogenesis as well as the cytotoxic mechanism of PirABvp. However, the quality of this MS is not good and needs to be improved.

Specific comments are as follows:

(Page 1, line 8)…AHPND was found to be a specific strain of bacteria that contains… Since AHPND was caused by specific strains of some Vibrio sp. (e.g. Vibrioparahaemolyticus and Vibrio harveyi) and a Shewanella sp. Strain TH2012 which contain pVA1 plasmid. Please specify Vibrio and Shewanella after “a specific strain of bacteria”. Some “VP” labels were added after PirAB as superscript, while some other “VP” labels were not. Please make them consistent. In my opinion, the terms of PirAB and PirABvp can be used to mention the PirAB toxins from Photorhabdus luminescens and Vibrio parahaemolyticus PirAB, respectively. To avoid misleading the readers, PirAB and PirABvp should be used more specifically in the MS. For instance, in Page 2, lines 43-49, the studies of binary PirAB toxins discussed here are in fact all related to Vibrioparahaemolyticus PirAB (i.e. PirABvp). Therefore, PirABvp will be more suitable in this paragraph.   (Page 2, lines 43-47) Among the binary PirAB toxin, ….. protein-ligand interaction [12,13]. Please add reference 3 here since this reference is the first report that described this hypothesis. When “Mab” and “Pab” terms were first used, please add “monoclonal antibody” and “polyclonal antibody” after them, respectively. (Figures) The information of scale bar is missing in all four figures. (Figure 2) In this figure, only one image (in different scales) was provided in individual experimental sets (i.e. negative control, 24 hr post-challenge…). To make this observation more solid, authors could add 1~2 more images for each sets in this figure. (Page 8, lines 183-185) “However, the present study showed that specificity of PirAB toxins is not very high, additionally the attachment is not organ, i.e., hepatopancreas dependent.” Please add references for this sentence. Most information of introduction and discussion is in duplicate (e.g. the first two paragraphs of introduction and discussion). Authors should reorganize these two sections to avoid repetition and add more information. For instance, do other pore-forming toxins also produce similar results as in this study?

Author Response

To,

The Reviewer,                                                                                    

Toxins

Manuscript " PirAB toxin binds to epithelial cells of the digestive tract and produce pathognomonic AHPND lesions in germ-free brine shrimp" Manuscript ID: toxins-658720

Dear Reviewer,

We sincerely thank you for your response and comments. We appreciate the thorough review and helpful suggestions, which we believe have contributed to improving the manuscript. We have considered and tried to address all the comments and suggestions given by you.

We have attempted to answer the comments systematically. In the following paragraphs, you will find our response to the comments.

(Page 1, line 8)…AHPND was found to be a specific strain of bacteria that contains… Since AHPND was caused by specific strains of some Vibrio sp. (e.g. Vibrio parahaemolyticus and Vibrio harveyi) and a Shewanella sp. Strain TH2012 which contain pVA1 plasmid. Please specify Vibrio and Shewanella after “a specific strain of bacteria”.

Taken account into reviewer comment, the sentence in the abstract accordingly changed (line 8-9 in the revised paper).

Some “VP” labels were added after PirAB as superscript, while some other “VP” labels were not. Please make them consistent. In my opinion, the terms of PirAB and PirABvp can be used to mention the PirAB toxins from Photorhabdus luminescens and Vibrio parahaemolyticus PirAB, respectively. To avoid misleading the readers, PirAB and PirABvp should be used more specifically in the MS. For instance, in Page 2, lines 43-49, the studies of binary PirAB toxins discussed here are in fact all related to Vibrio parahaemolyticus PirAB (i.e. PirABvp). Therefore, PirABvp will be more suitable in this paragraph. (Page 2, lines 43-47)

As suggested, we have carefully checked the entire manuscript and toxin name from Vibrio parahaemolyticus were changed to PirABvp and from Photorhabdus luminescens to PirAB

Among the binary PirAB toxin, ….. protein-ligand interaction [12,13]. Please add reference 3 here since this reference is the first report that described this hypothesis.

According to the suggestions, the reference [3] is added along with [12,13]

When “Mab” and “Pab” terms were first used, please add “monoclonal antibody” and “polyclonal antibody” after them, respectively. (Figures)

Taken account into reviewer comments, monoclonal antibody added after Mab and polyclonal antibody after Pab.

The information of scale bar is missing in all four figures.

As suggested, scale bar has been added in all figures

(Figure 2) In this figure, only one image (in different scales) was provided in individual experimental sets (i.e. negative control, 24 hr post-challenge…). To make this observation more solid, authors could add 1~2 more images for each sets in this figure.

According to the suggestions, additional figures (in figure 2) from 12 h control and 12 h treatments were added into the manuscript.

(Page 8, lines 183-185) “However, the present study showed that specificity of PirAB toxins is not very high, additionally the attachment is not organ, i.e., hepatopancreas dependent.” Please add references for this sentence.

Taken account into reviewer comment, we carefully described the finding in line 190-195 in the revised paper.

Most information of introduction and discussion is in duplicate (e.g. the first two paragraphs of introduction and discussion). Authors should reorganize these two sections to avoid repetition and add more information. For instance, do other pore-forming toxins also produce similar results as in this study?

As suggested, the discussion is reorganized to avoid the duplication

Round 2

Reviewer 2 Report

   This MS has been revised according to my previous suggestions. However, I still have a minor comment:

(Figures) Some white boxes can be found under scale bar. These boxes are not meaningful and should be removed.